# Learning with Social Influence through Interior Policy Differentiation

## Abstract

Animals develop novel skills not only through the interaction with the environment but also from the influence of the others. In this work we model the social influence into the scheme of reinforcement learning, enabling the agents to learn both from the environment and from their peers[1]. Specifically, we first define a metric to measure the distance between policies then quantitatively derive the definition of uniqueness. Unlike previous precarious joint optimization approaches, the social uniqueness motivation in our work is imposed as a constraint to encourage the agent to learn a policy different from the existing agents while still solve the primal task. The resulting algorithm, namely Interior Policy Differentiation (IPD), is able to learn a collection of policies that can solve a given task with distinct behaviors and brings about performance improvement as a byproduct in some cases.[2].

## 1 Introduction

The paradigm of Reinforcement Learning (RL), inspired by cognition and animal studies (Thorndike, 2017; Schultz et al., 1997), can be described as learning by interacting with the environment to maximize a cumulative reward (Sutton et al., 1998). From the perspective of ecology, biodiversity as well as the development of various skills are crucial to the continuation and evolution of species (Darwin, 1859; Pianka, 1970). Thus the behavioral diversity becomes a rising topic in RL. Previous works have tried to encourage the emergence of behavioral diversity in RL with two approaches: The first approach is to design interactive environments which contain sufficient richness and diversity. For example, Heess et al. (2017) show that rich environments enable agents to learn different locomotion skills even using the standard RL algorithms. Yet designing a complex environment requires manual efforts, and the diversity is limited by the obstacle classes. The second approach to increase behavioral diversity is to motivate agents to explore beyond just maximizing the reward for the given task. Zhang et al. (2019) proposed to maximize a heuristically defined novelty metric between policies through task-novelty joint optimization, but the final performance of agents is not guaranteed.

In this work, we address the topic of *policy differentiation* in RL, i.e., to improve the diversity of RL agents while keeping their ability to solve the primal task. We draw the inspiration from the Social Influence in animal society (Rogoff, 1990; Ryan & Deci, 2000; van Schaik & Burkart, 2011; Henrich, 2017; Harari, 2014) and formulate the concept of social influence in the reinforcement learning paradigm. Our learning scheme is illustrated in Fig 1. The target agent not only learns to interact with the environment to maximize the reward but also differentiate the actions it takes in order to be different from other existing agents.

Since the social influence often acts on people passively as a sort of peer pressure, we implement the social influence in terms of social uniqueness motivation (Chan et al., 2012) and consider it as a constrained optimization problem. In the following of our work, we first define a rigorous policy distance metric in the policy space to compare the similarity of the agents. Then we develop an optimization constraint using the proposed metric, which brings immediate rather than episodic feedback in the learning process. A novel method, namely Interior Policy Differentiation (IPD), is further

---

[1] In this work, we use the term "peer" to denote a population of RL agents.

[2] Code will be made available soon

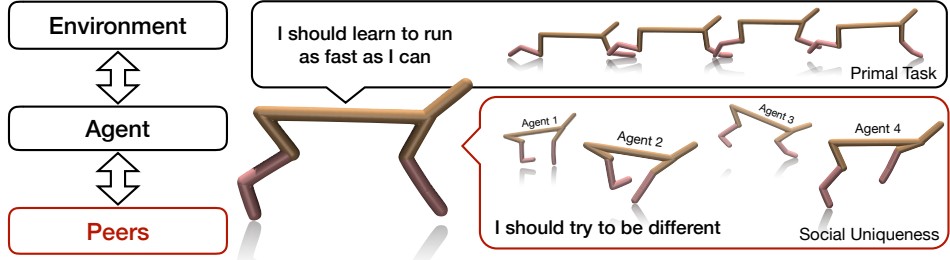

Figure 1: The illustration of learning with social influence. Instead of focusing only on the primal task, an additional constraint is introduced to the target agent, motivating it to not only perform well in the primal task but also take actions differently to other existing agents.

proposed as a better solution for the constrained policy optimization problem. We benchmark our method on several locomotion tasks and show it can learn various diverse and well-behaved policies for the given tasks based on the standard Proximal Policy Optimization (PPO) algorithm (Schulman et al., 2017).

## 2    RELATED WORK

**Intrinsic motivation methods.** The Variational Information Maximizing Exploration (VIME) method is designed by Houthooft et al. (2016) to tackle the sparse reward problems. In VIME, an intrinsic reward term based on the maximization of information gains is added to contemporary RL algorithms to encourage exploration. The curiosity-driven methods, proposed by Pathak et al. (2017) and Burda et al. (2018a) define intrinsic rewards according to prediction errors of neural networks. i.e., when taking previous unseen states as inputs, networks trained with previous states will tend to predict with low accuracy, so that such prediction errors can be viewed as rewards. Burda et al. (2018b) proposed Random Network Distillation (RND) to quantify intrinsic reward by prediction differences between a fixed random initialized network and another randomly initialized network trained with previous state information. Liu et al. (2019) proposed Competitive Experience Replay (CER), in which they use two actors and a centralized critic, and defined an intrinsic reward by the state coincidence of two actors. The values of intrinsic rewards are fixed to be $\pm 1$ for the two actors separately. All of those approaches leverage the weighted sum of the external rewards, i.e., the primal rewards provided by environments, and intrinsic rewards that provided by different heuristics. A challenging problem is the trade-off between external rewards and intrinsic rewards. The Task-Novelty Bisector (TNB) learning method introduced by Zhang et al. (2019) aims to solve such problem by jointly optimize the extrinsic rewards and intrinsic rewards. Specifically, TNB updates the policy in the direction of the angular bisector of the two gradients, i.e., gradients of the extrinsic and intrinsic objective functions. However, the foundation of such joint optimization is not solid. Besides, creating an extra intrinsic reward function and evaluating the novelty of states or policies always requires additional neural networks such as auto-encoders. Thus extra computation expenses are needed (Zhang et al., 2019) .

**Diverse behaviors from rich environments and algorithms.** Heess et al. (2017) introduce the Distributed Proximal Policy Optimization (DPPO) method and enable agents with simulated bodies to learn complex locomotion skills in a diverse set of challenging environments. Although the learning reward they utilize is straightforward, the skills their policy learned are quite impressive and effective in traveling terrains and obstacles. Their work shows that rich environments can encourage the emergence of different locomotion behaviors, but extra manual efforts are required in designing such environments. The research of Such et al. (2018) shows that different RL algorithms may converge to different policies for the same task. The authors find that algorithms based on policy gradient tend to converge to the same local optimum in the game of Pitfall, while off-policy and value-based algorithms are prone to learn sophisticated strategies. On the contrary, in this paper, we are more interested in how to learn different policies through a single learning algorithm and learn the capability of avoiding local optimum.

**Population-based novelty-seeking methods.** Pugh et al. (2016) establish a standard framework for understanding and comparing different approaches to searching for quality diversity (QD). Conti et al. (2018) investigate adding novelty search (NS) and QD to evolution strategies (ES) to avoid local optima as well as achieve higher performance. Lehman & Stanley (2011; 2008) conclude that deriving an open-ended search algorithm that operates without pressure towards the ultimate objective is possible, suggesting ignoring the objective may often benefit the search itself. The work of Wang et al. (2019) yields a new kind of open-ended algorithm which indicates the solution to one environment might be a stepping stone to a new level of performance in another. Such et al. (2017) evolve a DNN with a population-based genetic algorithm (GA) for challenging RL tasks. By improving the vanilla TRPO algorithm (Schulman et al., 2015), Kurutach et al. (2018) maintains model uncertainty given the data collected from the environment via an ensemble of deep neural networks.

## 3 QUANTIFYING THE DISTANCE BETWEEN POLICIES

To encourage the emergence of behavioral diversity in RL, we first define a metric to measure the difference between policies, which is the foundation for the later algorithm we propose. We denote the learned policies as $\{\pi_{\theta_i}; \theta_i \in \Theta, i = 1, 2, ...\}$, wherein $\theta_i$ represents parameters of the $i$-th policy, $\Theta$ denotes the whole parameter space. In the following, we omit $\pi$ and denote a policy $\pi_{\theta_i}$ as $\theta_i$ for simplicity unless stated otherwise.

### 3.1 DEFINITION

Mathematically, a metric should satisfy three important properties, namely the identity, the symmetry as well as the triangle inequality.

**Definition 1** *A metric space is an ordered pair $(M, d)$ where $M$ is a set and $d$ is a metric on $M$, i.e., a function $d \colon M \times M \to \mathbb{R}$ such that for any $x, y, z \in M$, the following holds:*
*1.    $d(x, y) \geq 0, d(x, y) = 0 \Leftrightarrow x = y$,*
*2.    $d(x, y) = d(y, x)$,*
*3.    $d(x, z) \leq d(x, y) + d(y, z)$.*

We use the Total Variance Divergence $D_{TV}$ (Schulman et al., 2015) to measure the distance between policies. Concretely, for discrete probability distributions $p$ and $q$, this distance is defined as $D_{TV}(p, q) = \sum_i |p_i - q_i|$. [3][4]

**Theorem 1 (Metric Space $(\Theta, \overline{D}^\rho_{TV})$)** *The expectation of $D_{TV}(\cdot, \cdot)$ of two policies over any state distribution $\rho(s)$:*

$$\overline{D}^\rho_{TV}(\theta_i, \theta_j) := \mathbb{E}_{s \sim \rho(s)}[D_{TV}(\theta_i(s), \theta_j(s))], \tag{1}$$

*is a metric on $\Theta$, thus $(\Theta, \overline{D}^\rho_{TV})$ is a metric space.*

The proof of Theorem 1 is in Appendix A. It is worth mentioning that, although TVD is used in our work, we can easily extend the result to use other distance between distributions as substitutes of TVD (e.g. Jensen Shannon divergence $D_{JS}$ or Wasserstein metric $D_W$) (Endres & Schindelin, 2003; Fuglede & Topsoe, 2004; Villani, 2008), and similar results can be get

**Corollary 1** *Let $\overline{D}^\rho_{JS} := \mathbb{E}_{s \sim \rho(s)}[D_{JS}(\theta_i(s), \theta_j(s))]$ and $\overline{D}^\rho_W := \mathbb{E}_{s \sim \rho(s)}[D_W(\theta_i(s), \theta_j(s))]$, $(\Theta, \overline{D}^\rho_{JS})$ and $(\Theta, \overline{D}^\rho_W)$ are also metric spaces.*

On top of the metric space $(\Theta, \overline{D}^\rho_{TV})$, we could then compute the uniqueness of a policy.

**Definition 2 (Uniqueness of Policy)** *Given a reference policy set $\Theta_{ref}$ such that $\Theta_{ref} = \{\theta_i^{ref}, i = 1, 2, ...\}, \Theta_{ref} \subset \Theta$, the uniqueness $U(\theta | \Theta_{ref})$ of policy $\theta$ is the minimal difference between $\theta$ and all policy in the reference policy set, i.e.,*

$$U(\theta | \Theta_{ref}) := \min_{\theta_j \in \Theta_{ref}} \overline{D}^\rho_{TV}(\theta, \theta_j). \tag{2}$$

---

[3]It can be extended to continuous state and action spaces by replacing the sums with integrals.
[4]The factor $\frac{1}{2}$ in Schulman et al. (2015) is omitted in our work for conciseness.

Consequently, to motivate RL with the social uniqueness, we hope our method can maximize the uniqueness of a new policy, i.e., $\max_\theta U(\theta|\Theta_{ref})$, where the $\Theta_{ref}$ includes all the existing policies.

## 3.2 Estimation of $\overline{D}^\rho_{TV}(\theta_i, \theta_j)$

In practice, the calculation of $\overline{D}^\rho_{TV}(\theta_i, \theta_j)$ is based on Monte Carlo estimation. i.e., we need to sample $s$ from $\rho(s)$. Although in finite state space we can get precise estimation after establishing ergodicity, problem arises when we are facing continuous state cases. i.e. it is difficult to efficiently get enough samples.

Formally, we denote the domain of $\rho(s)$ as $\mathcal{S}$ and denote the domain of $\rho_\theta(s)$ as $\mathcal{S}_\theta \subset \mathcal{S}$, where $\rho_\theta(s) := \rho(s|s \sim \theta)$ and in finite time horizon problems $\rho(s|s \sim \theta) = P(s_0 = s|\theta) + P(s_1 = s|\theta) + ... + P(s_T = s|\theta)$. As we only care about the reachable regions, the domain $\mathcal{S}$ can be divided by $\mathcal{S} = \lim_{N \to \infty} \bigcup_{i=1}^N \mathcal{S}_{\theta_i}$.

In order to improve the sample efficiency, we propose to approximate $\overline{D}^\rho_{TV}(\theta_i, \theta_j)$ with $\overline{D}^{\rho_\theta}_{TV}(\theta_i, \theta_j)$, where $\theta$ is a certain fixed behavior policy that irrelevant to $\theta_i, \theta_j$. Such approximation requires a necessary condition:

**Condition 1** *The domain of possible states are similar between different policies:*
$$\sum_{s \in \mathcal{S}} P(s \in (\mathcal{S}_\theta \cup \mathcal{S}_{\theta_j}) \setminus (\mathcal{S}_\theta \cap \mathcal{S}_{\theta_j})) \ll \sum_{s \in \mathcal{S}} P(s \in (\mathcal{S}_\theta \cap \mathcal{S}_{\theta_j})), \forall j. \tag{3}$$

When such condition holds, we can use $\rho(s|s \sim \theta)$ as our choice of $\rho(s)$, and the properties in Definition 1 still holds.

In practice, the Condition 1 always holds as we can ensure this by adding sufficiently large noise on $\theta$, while the permitted state space is always limited. And for more general cases, to satisfy the properties in Definition 1, we must sample $s$ from $\mathcal{S}_\theta \cup \mathcal{S}_{\theta_j}$, accordingly,

$$\begin{aligned} \overline{D}^\rho_{TV}(\theta, \theta_j) &= \mathbb{E}_{s \sim (\mathcal{S}_\theta \cup \mathcal{S}_j)}[D_{TV}(\theta(s), \theta_j(s))] \\ &= \mathbb{E}_{s \sim (\mathcal{S}_\theta \cap \mathcal{S}_{\theta_j})}[D_{TV}(\theta(s), \theta_j(s))] + \mathbb{E}_{s \sim (\mathcal{S}_\theta \cup \mathcal{S}_{\theta_j}) \setminus \mathcal{S}_{\theta_j}}[D_{TV}(\theta(s), \mathcal{N})] + \\ &\quad \mathbb{E}_{s \sim (\mathcal{S}_\theta \cup \mathcal{S}_{\theta_j}) \setminus \mathcal{S}_\theta}[D_{TV}(\mathcal{N}, \theta_j(s))] \end{aligned} \tag{4}$$

where $\mathcal{N}$ represents random action when a policy have never been trained or visited such state domain. Plugging Eq.(4) into Eq.(2), the objective function of policy differentiation is

$$\begin{aligned} \max_\theta \min_{\theta_j \in \Theta_{ref}} \overline{D}^\rho_{TV}(\theta, \theta_j) &= \mathbb{E}_{s \sim (\mathcal{S}_\theta \cap \mathcal{S}_{\theta_j})}[D_{TV}(\theta(s), \theta_j(s))] \\ &+ \mathbb{E}_{s \sim (\mathcal{S}_\theta \cup \mathcal{S}_{\theta_j}) \setminus \mathcal{S}_{\theta_j}}[D_{TV}(\theta(s), \mathcal{N})] + \mathbb{E}_{s \sim (\mathcal{S}_\theta \cup \mathcal{S}_{\theta_j}) \setminus \mathcal{S}_\theta}[D_{TV}(\mathcal{N}, \theta_j(s))] \end{aligned} \tag{5}$$

While the first two terms are related to the policy $\theta$, the last term is only related to the domain $\mathcal{S}_\theta$. If we enable sufficient exploration in training as well as in the initialization of $\theta$, the last term will disappear (i.e. $\mathcal{S}_{\theta_j} \subset \mathcal{S}_\theta$). Hence we can also use $\overline{D}^{\rho_{\theta_i}}_{TV}(\theta_i, \theta_j)$ as an approximation of $\overline{D}^\rho_{TV}(\theta_i, \theta_j)$ in training of $\theta_i$ as long as sufficient exploration is guaranteed.

**Proposition 1 (Unbiased Single Trajectory Estimation)** *The estimation of $\rho_\theta(s)$ using a single trajectory $\tau$ is unbiased.*

The proof of Proposition 1 is in Appendix B. Given the definition of uniqueness and a practically unbiased sampling method, the next step is to develop an efficient learning algorithm.

## 4 Interior Policy Differentiation

In the traditional RL paradigm, maximizing the expectation of cumulative rewards $g = \sum_{t=0} \gamma^t r_t$ is commonly used as the objective. i.e. $\max_{\theta \in \Theta} \mathbb{E}_{\tau \sim \theta}[g]$, where $\tau \sim \theta$ denotes a trajectory $\tau$ sampled from the policy $\theta$ using Monte Carlo methods.

To improve the behavioral diversity of different agents, the learning objective must take both reward from the primal task and the policy uniqueness into consideration. Previous approaches (Houthooft et al., 2016; Pathak et al., 2017; Burda et al., 2018a;b; Liu et al., 2019) often directly write the weighted sum of the reward from the primal task and the intrinsic reward $g_{\text{int}} = \sum_{t=0} \gamma^t r_{\text{int},t}$, where $r_{\text{int},t}$ denotes the *intrinsic reward* (e.g., $r_{\text{int}} = \min_{\theta_j \in \Theta_{ref}} \overline{D}^\rho_{TV}(\theta, \theta_j)$ as the uniqueness reward in our case) as follows,

$$\max_{\theta \in \Theta} \quad \mathbb{E}_{\tau \sim \theta}[g_{\text{total}}] = \max_{\theta \in \Theta} \quad \mathbb{E}_{\tau \sim \theta}[\alpha \cdot g_{\text{task}} + (1 - \alpha) \cdot g_{\text{int}}], \tag{6}$$

where $0 < \alpha < 1$ is a weight parameter. Such an objective is sensitive to the selection of $\alpha$ as well as the formulation of $r_{\text{int}}$. For example, in our case formulating the intrinsic reward $r_{\text{int}}$ as $\min_{\theta_j} \overline{D}^\rho_{TV}(\theta, \theta_j)$, $\exp[\min_{\theta_j} \overline{D}^\rho_{TV}(\theta, \theta_j)]$ and $-\exp[-\min_{\theta_j} \overline{D}^\rho_{TV}(\theta, \theta_j)]$ will result in significantly different results. Besides, a trade-off arises in the selection of $\alpha$: while a large $\alpha$ may undermine the contribution of intrinsic reward, a small $\alpha$ could ignore the importance of the reward, leading to the failure of agent in solving the primal task.

To tackle these issues, we draw inspiration from the observation that social uniqueness motivates people in passive ways. In other words, it plays more like a constraint rather than an additional target. Therefore, we change the multi-objective optimization problem in Eq.(6) into a constrained optimization problem as:

$$\begin{aligned} \max_{\theta \in \Theta} \quad & \mathbb{E}_{\tau \sim \theta}[g_{\text{task}}], \\ s.t. \quad & \overline{r}_{\text{int},t} - r_0 \geq 0, \forall t = 1, 2, ..., T, \end{aligned} \tag{7}$$

where $r_0$ is a threshold indicating minimal permitted uniqueness, and $\overline{r}_{\text{int},t}$ denotes a moving average of $r_{\text{int},t}$. Further discussion on the selection of $r_0$ will be deliberated in Appendix D.

From the perspective of optimization, Eq.(6) can be viewed as a penalty method which replaces the constrained optimization problem in Eq.(7) with the penalty term $r_{\text{int}}$ and the penalty coefficient $\frac{1-\alpha}{\alpha} > 0$, where the difficulty lies in the selection of $\alpha$. The work of Zhang et al. (2019)) tackles this challenge by the Task Novel Bisector (TNB) in the form of Feasible Direction Methods (FDMs) (Zoutendijk, 1960). As a heuristic approximation, that approach requires reward shaping and intensive emphasis on $r_{\text{int},t}$. Instead, in this work we propose to solve the constrained optimization problem Eq.(7) by resembling the Interior Point Methods (IPMs) (Potra & Wright, 2000; Dantzig & Thapa, 2006). In vanilla IPMs, the constrained optimization problem in Eq.(7) is solved by reforming it to an unconstrained form with an additional barrier term in the objective as

$$\max_{\theta \in \Theta} \quad \mathbb{E}_{\tau \sim \theta}[g_{\text{task}} + \sum_{t=0}^{T} \alpha \log(r_{\text{int},t} - r_0)]. \tag{8}$$

The limit of Eq.(8) when $\alpha \to 0$ then leads to the solution of Eq.(7). Readers please refer to Appendix G for more discussion on the correspondence between those novel policy seeking methods and constrained optimization methods.

However, directly applying the IPMs is computationally challenging and numerically unstable, especially when $\alpha$ is small. Luckily, in our proposed RL paradigm where the behavior of an agent is influenced by its peers, a more natural way can be used. Precisely, since the learning process is based on sampled transitions, we can simply bound the collected transitions in the feasible region by permitting previous trained $M$ policies $\theta_i \in \Theta_{\text{ref}}, i = 1, 2, ..., M$ sending termination signals during the training process of new agents. In other words, we implicitly bound the feasible region by terminating any new agent that steps outside it. Consequently, during the training process, all valid samples we collected are inside the feasible region, which means these samples are less likely to appear in previously trained policies. At the end of the training, we then naturally obtain a new policy that has sufficient uniqueness. In this way, we no longer need to consider the trade-off problem between intrinsic and extrinsic rewards deliberately. The learning process of our method is thus more robust and no longer suffer from objective inconsistency. As our formulation of the constrained optimization problem Eq.(7) is inspired by IPMs, we name our approach as Interior Policy Differentiation (IPD) method.

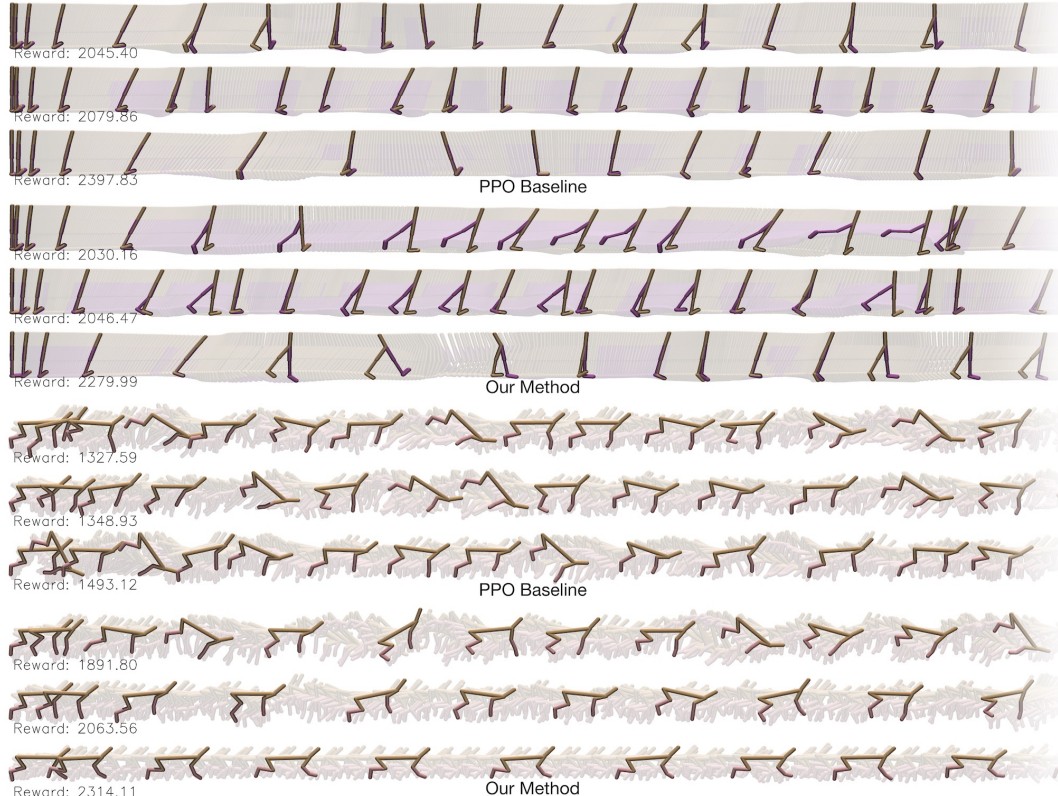

Figure 2: Results of policy differentiation on Walker2d-v3 and HalfCheetah-v3. Compared to the PPO baseline, our method significantly diversifies trained policies while maintaining their performances.

# 5 EXPERIMENTS

**The MuJoCo environment** We demonstrate our proposed method on the OpenAI Gym where the physics engine is based on MuJoCo (Brockman et al., 2016; Todorov et al., 2012). Concretely, we test on three locomotion environments, the Hopper-v3 (11 observations and 3 actions), Walker2d-v3 (11 observations and 2 actions), and HalfCheetah-v3 (17 observations and 6 actions). In our experiments, all the environment parameters are set as default values.

**Uniqueness beyond intrinsic stochasticity** Experiments in Henderson et al. (2018) show that policies that perform differently can be produced by simply selecting different random seeds before training. Before applying our method to improve behavior diversity, we firstly benchmark how much uniqueness can be generated from the stochasticity in the training process of vanilla RL algorithms as well as the random weight initialization. In this work, we mainly demonstrate our proposed method based on PPO(Schulman et al., 2017). The extension to other popular algorithms is straightforward. We also compare our proposed method with the TNB and weighted sum reward (WSR) approaches as different ways to combine the goal of the task and the uniqueness motivation (Zhang et al., 2019). More implementation details are depicted in Appendix D.

## 5.1 UNIQUENESS AND PERFORMANCE COMPARISON

According to Theorem 2, the uniqueness $r_{int}$ in equation (7) under our uniqueness metric can be unbiased approximated by $r_{\text{int}} = \min_{\theta_j \in \Theta_{ref}} \overline{D}_{TV}^{\rho_\theta}(\theta(s_t), \theta_j(s_t))$. i.e., we utilize the metric directly in learning new policies instead of applying any kind of reshaping.

We implement WSR, TNB, and our method in the same experimental settings and for each method, 10 different policies are trained and try to be unique with regard to all previously trained policies

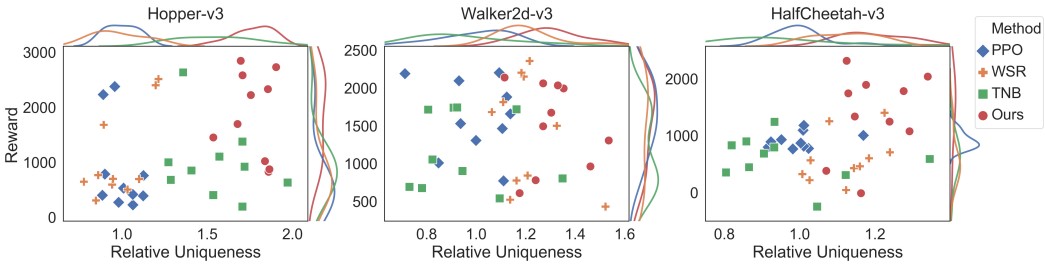

Figure 3: The scatter plot of different policies in terms of Uniqueness and Performance in Hopper-v3, Walker2d-v3 and HalfCheetah-v3 environments. The value of uniqueness is normalized to relative uniqueness by regarding the averaged uniqueness of PPO policies as the baseline.

Table 1: The reward and success rate of learned 10 policies using different methods

| Method | Reward | | | Success Rate | | |
|---|---|---|---|---|---|---|
| | Hopper | Walker2d | HalfCheetah | Hopper | Walker2d | HalfCheetah |
| PPO | $839 \pm 753$ | $\mathbf{1611 \pm 467}$ | $913 \pm 134$ | 1.0 | 1.0 | 0.7 |
| PPO+WSR | $1083 \pm 768$ | $1429 \pm 692$ | $603 \pm 407$ | 1.0 | 0.7 | 0.4 |
| PPO+TNB | $1064 \pm 644$ | $1160 \pm 484$ | $592 \pm 384$ | 1.0 | 0.9 | 0.5 |
| PPO+Ours | $\mathbf{1858 \pm 744}$ | $1506 \pm 541$ | $\mathbf{1442 \pm 588}$ | 1.0 | 1.0 | 0.9 |

sequentially. Concretely, the $1st$ policy is trained by ordinary PPO without any social influence. The $2nd$ policy should be different from $1st$ policy, and the $3rd$ should be different from the previous two policies, and so on. Fig.2 shows the qualitative results of our method. We visualize the motion of agents by drawing multiple frames representing the pose of agents at different time steps in the same row. The horizontal interval between consecutive frames is proportional to the velocity of agents. The settings of the frequency of highlighted frames and the correlation between interval and velocity are fixed for each environment. The visualization starts from the beginning of each episode and therefore the readers can get sense of the process of acceleration as well as the pattern of motion of agents clearly.

Fig. 3 shows our experimental results in terms of uniqueness (the x-axis) and the performance (the y-axis). Policies in the upper right are the more unique ones with higher performance. In Hopper and HalfCheetah, our proposed method distinctively outperforms other methods. In Walker2d, both WSR and our method work well in improving the uniqueness of policies, but none of the three methods can find way to surpass the performance of PPO apparently. Detailed comparison on the task related rewards are carried out in Table 1. A box figure depicting the performance of each trained policy and their reward gaining curve are disposed in Fig.5 and Fig.6 in Appendix C. And Fig.7 in Appendix C provides more detailed results from the view of uniqueness.

## 5.2  SUCCESS RATE OF EACH METHOD

In addition to averaged reward, we also use success rate as another metrics to compare the performance of different approaches. In this work, we consider a policy is success when its performance is at least as good as the averaged performance of policies trained without social influences. To be specific, we use the averaged final performance of PPO as the baseline. If a new policy, which aims at performing differently to solve the same task, surpasses the baseline during its training process, it will be regarded as a successful policy. Through the success rate, we know the policy does not learn unique behavior at the expense of performance. Table 1 shows the success rate of all the methods, including the PPO baseline. The results show that our method can always surpass the average baseline during training. Thus the performance of our method can always be insured.

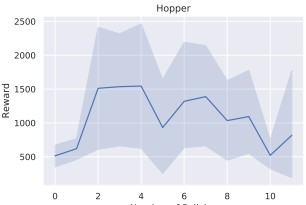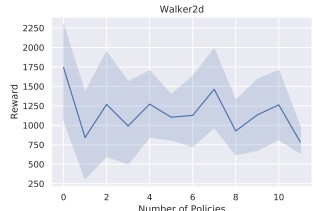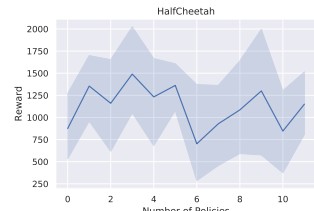

Figure 4: Performance curves along with the number of peers. The results are averaged over 5 repetitions of individual experiments.

## 5.3 BETTER POLICY DISCOVERY

In our experiments, we observed noticeable performance improvements in the Hopper and the HalfCheetah environments. For the environment of Hopper, in many cases, the agents trained with PPO tend to learn a policy that jumps as far as possible and then fall to the ground and terminate this episode (please refer to Fig.11 in Appendix E). Our proposed method can prevent new policies from always falling into the same local minimum. After the first policy being trapped in a local minimum, the following policies will try other approaches to avoid the same behavior, explore other feasible action patterns, and thereafter the performance may get improved. Such property shows that our method can be a helpful enhancement of the traditional RL scheme, which can be epitomized as policies could make mistakes, but they should explore more instead of hanging around the same local minimum. The similar feature attributes to the reward growth in the environment of HalfCheetah.

Moreover, we can illuminate the performance improvement of HalfCheetah from another perspective. The environment of HalfCheetah is quite different from the other two for there is no explicit termination signal in its default settings (i.e., no explicit action like falling to the ground would trigger termination). At the beginning of the learning process, an agent will act randomly, resulting in massive repeat, trivial samples as well as large control costs. In our learning scheme, since the agent also interacts with the peers, it can receive termination signals from the peers to prevent wasting too much effort acting randomly. During the learning process in our method, an agent will first learn to terminate itself as soon as possible to avoid heavy control costs by imitating previous policies and then learns to behave differently to pursue higher reward. From this point of view, such learning process can be regarded as a kind of implicit curriculum.

## 5.4 SCALE OF THE INFLUENCE

As the number of policies learned with social influence grows, the difficulty of finding a unique policy may also increase. Later policies must keep away from all previous solutions. The results of our ablation study on how the performance changes under different scales of social influence (i.e., the number of peers) is shown in Fig. 4, where the thresholds are selected according to our previous ablation study in Sec. D. The performance decrease is more obvious in Hopper than the other two environments for the action space of Hopper is only 3 dimensional. Thus the number of possible diverse policies can be discovered is limited.

## 6 CONCLUSION

In this work, we develop an efficient approach to motivate RL to learn diverse strategies inspired by social influence. After defining the distance between policies, we introduce the definition of policy uniqueness. Regarding the problem as constrained optimization problem, our proposed method, Interior Policy Differentiation (IPD), draws the key insight of the Interior Point Methods. And our experimental results demonstrate IPD can learn various well-behaved policies, and our approach can help agents to avoid local minimum and can be interpreted as a kind of implicit curriculum learning in certain cases.

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

## A    PROOF OF THEOREM 1

The first two properties are obviously guaranteed by $\overline{D}_{TV}^{\rho}$. As for the triangle inequality,

$$
\begin{aligned}
\mathbb{E}_{s \sim \rho(s)}[D_{TV}(\theta_i(s), \theta_k(s)] &= \mathbb{E}_{s \sim \rho(s)}[\sum_{l=1}^{|\mathcal{A}|} |\theta_i(s) - \theta_k(s)|] \\
&= \mathbb{E}_{s \sim \rho(s)}[\sum_{l=1}^{|\mathcal{A}|} |\theta_i(s) - \theta_j(s) + \theta_j(s) - \theta_k(s)|] \\
&\leq \mathbb{E}_{s \sim \rho(s)}[\sum_{l=1}^{(|\mathcal{A}|} |\theta_i(s) - \theta_j(s)| + |\theta_j(s) - \theta_k(s)|)] \\
&= \mathbb{E}_{s \sim \rho(s)}[\sum_{l=1}^{|\mathcal{A}|} |\theta_i(s) - \theta_j(s)|] + \mathbb{E}_{s \sim \rho(s)}[\sum_{l=1}^{|\mathcal{A}|} |\theta_j(s) - \theta_k(s)|] \\
&= \mathbb{E}_{s \sim \rho(s)}[D_{TV}(\theta_i(s), \theta_j(s)] + \mathbb{E}_{s \sim \rho(s)}[D_{TV}(\theta_j(s), \theta_k(s)]
\end{aligned}
$$

## B    PROOF OF PROPOSITION 1

$$
\begin{aligned}
\rho_\theta(s) &= P(s_0 = s|\theta) + P(s_1 = s|\theta) + ... + P(s_T = s|\theta) \\
&\stackrel{L.L.N.}{=} \lim_{N \to \infty} \frac{\sum_{i=1}^{N} I(s_0 = s|\tau_i)}{N} + \frac{\sum_{i=1}^{N} I(s_1 = s|\tau_i)}{N} + ... + \frac{\sum_{i=1}^{N} I(s_T = s|\tau_i)}{N} \\
&= \lim_{N \to \infty} \frac{\sum_{j=0}^{T} \sum_{i=1}^{N} I(s_j = s|\tau_i)}{N} \\
\overline{\rho}_\theta(s) &= \sum_{i=1}^{N} \sum_{j=0}^{T} \frac{I(s_j = s|\tau_i)}{N} \\
\mathbb{E}[\overline{\rho}_\theta(s) - \rho_\theta(s)] &= 0
\end{aligned}
$$

## C  DETAILS OF UNIQUENESS AND PERFORMANCE

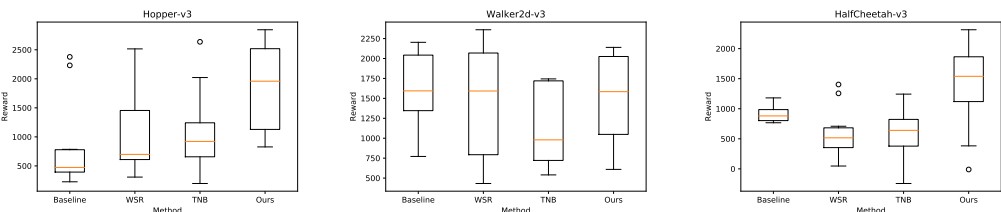

Figure 5: The performance of different methods in the Hopper, Walker and HalfCheetah environments. The results are collected from 10 learned policies based on PPO. The box extends from the lower to upper quartile values of the data, with a line at the median. The whiskers extend from the box to show the range of the data. Flier points are those past the end of the whiskers.

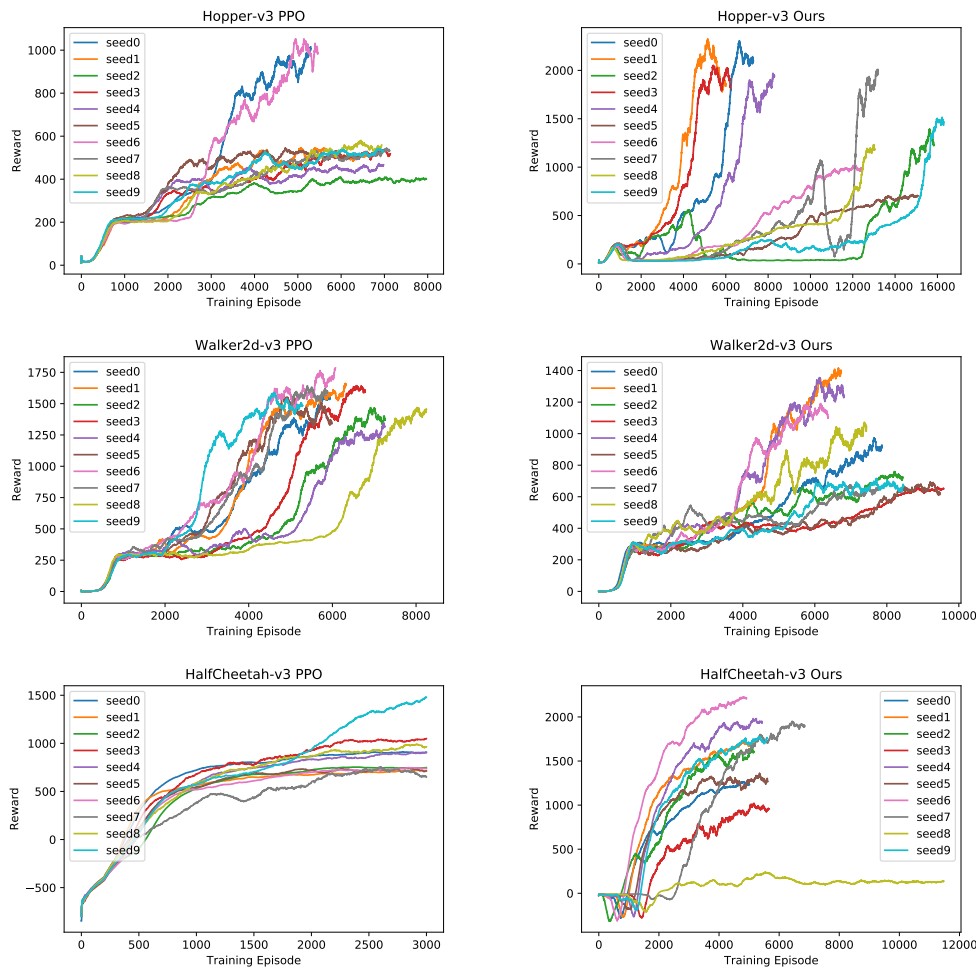

Figure 6: Avoid local minimum in Hopper and HalfCheetah: the left two figures show 10 policies generated by PPO in each environment, the right two figures show 10 policies generated by our method in each environment

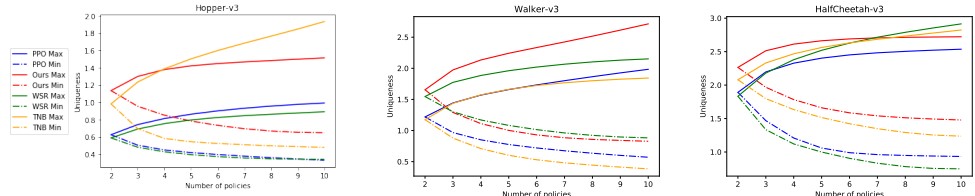

Figure 7: Maximal and minimal between policy uniqueness in Hopper, Walker2d and HalfCheetah environments. The results are averaged over all possible combinations of 10 policies. As TNB and WSR optimize the uniqueness reward directly, their uniqueness sometimes can exceed our proposed method. However, such direct optimization will lead to decreasing in task related performance as cost. To tackle the trade-off problem, carefully hyper-parameter tuning and reward shaping is always a must. Detailed comparison on the task related rewards are carried out in Table 1

## D    IMPLEMENTATION DETAILS

**Calculation of** $D_{TV}$    We use deterministic part of policies in the calculation of $D_{TV}$, i.e., we remove the Gaussian noise on the action space in PPO and use $D_{TV}(a_1, a_2) = |a_1 - a_2|$.

**Network Structure**    We use MLP with 2 hidden layers as our actor models in PPO. The first hidden layer is fixed to have 32 units. Our ablation study on the choice of unit number in the second layer is detailed in Table.2, Table3 and Fig.8. Moreover, we choose to use 10, 64 and 256 hidden units for the three tasks respectively in all of the main experiments, after taking the success rate (Table.2), performance (Table.3) and computation expense (i.e. the preference to use less unit when the other two factors are similar) into consideration.

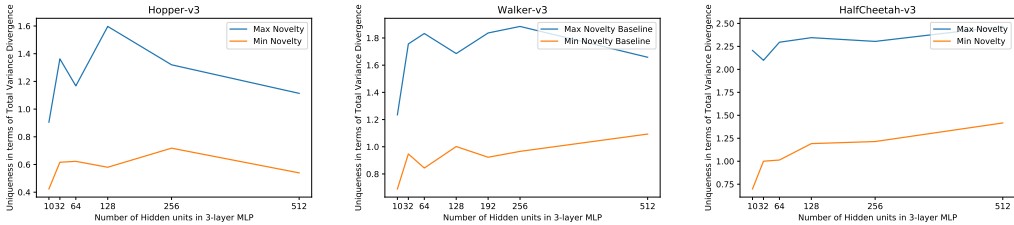

Figure 8: Uniqueness increases a little as network complexity rises

Table 2: The success rate of learned 10 novel policies by our method in different environments under different thresholds

| Environment | Threshold | 10 hidden | 64 hidden | 256 hidden | 512 hidden |
|---|---|---|---|---|---|
| Hopper | 0.5 | 0.7 | 0.8 | 0.7 | — |
| | 0.6 | 1.0 | 0.7 | 1.0 | — |
| | 0.7 | 0.9 | 0.4 | 0.7 | — |
| | 0.8 | 0.5 | 0.5 | 0.3 | — |
| | 0.9 | 0.3 | 0.1 | 0.4 | — |
| Walker2d | 1.0 | 0.2 | 0.8 | 1.0 | — |
| | 1.1 | 0.3 | 0.8 | 1.0 | — |
| | 1.2 | 0.1 | 0.9 | 1.0 | — |
| | 1.3 | 0.2 | 0.8 | 0.8 | — |
| | 1.4 | 0.1 | 0.6 | 1.0 | — |
| HalfCheetah | 1.1 | — | 0.3 | 0.9 | 0.8 |
| | 1.2 | — | 0.8 | 0.8 | 0.7 |
| | 1.3 | — | 1.0 | 1.0 | 1.0 |
| | 1.4 | — | 0.4 | 0.9 | 1.0 |
| | 1.5 | — | 0.2 | 0.2 | 0.7 |

**Training Timesteps**    We fix the training timesteps in our experiments. The timesteps are fixed to be 1M in Hopper-v3, 1.6M for Walker2d-v3 and 3M for HalfCheetah.

Table 3: The final training performance of learned 10 novel policies by our method in different environments under different thresholds, for Hopper and Walker, $h_1 = 10, h_2 = 64, h_3 = 256$; for HalfCheetah, $h_1 = 64, h_2 = 256, h_3 = 512$

| Environment | Threshold | Average Performance | | | Top 30% Performance | | |
|---|---|---|---|---|---|---|---|
| hid num | | $h_1$ | $h_2$ | $h_3$ | $h_1$ | $h_2$ | $h_3$ |
| Hopper | 0 | $450 \pm 135$ | $914 \pm 735$ | $704 \pm 598$ | $607 \pm 30$ | $1665 \pm 726$ | $1202 \pm 737$ |
| | 0.6 | $\mathbf{1858 \pm 744}$ | $1040 \pm 914$ | $1626 \pm 869$ | $\mathbf{2719 \pm 108}$ | $\mathbf{2263 \pm 764}$ | $\mathbf{2628 \pm 207}$ |
| | 0.7 | $1180 \pm 740$ | $593 \pm 159$ | $785 \pm 462$ | $2188 \pm 510$ | $769 \pm 17$ | $1287 \pm 590$ |
| | 0.8 | $397 \pm 283$ | $767 \pm 743$ | $950 \pm 929$ | $744 \pm 152$ | $1594 \pm 875$ | $2129 \pm 896$ |
| | 0.9 | $154 \pm 142$ | $235 \pm 279$ | $604 \pm 554$ | $335 \pm 122$ | $645 \pm 113$ | $1347 \pm 387$ |
| | 1.0 | $187 \pm 229$ | $298 \pm 256$ | $499 \pm 754$ | $490 \pm 194$ | $648 \pm 141$ | $1294 \pm 979$ |
| Walker | 0 | $\mathbf{1611 \pm 467}$ | $1504 \pm 502$ | $\mathbf{1724 \pm 584}$ | $\mathbf{2163 \pm 48}$ | $2018 \pm 54$ | $\mathbf{2311 \pm 45}$ |
| | 1.0 | $725 \pm 487$ | $1174 \pm 599$ | $1571 \pm 692$ | $1346 \pm 468$ | $2042 \pm 77$ | $2270 \pm 37$ |
| | 1.1 | $725 \pm 654$ | $\mathbf{1506 \pm 541}$ | $1453 \pm 480$ | $1561 \pm 598$ | $\mathbf{2079 \pm 44}$ | $1903 \pm 104$ |
| | 1.2 | $487 \pm 375$ | $1061 \pm 346$ | $1211 \pm 657$ | $880 \pm 451$ | $1552 \pm 114$ | $2114 \pm 47$ |
| | 1.3 | $405 \pm 647$ | $1138 \pm 591$ | $995 \pm 420$ | $1124 \pm 795$ | $1984 \pm 197$ | $1523 \pm 313$ |
| | 1.4 | $393 \pm 518$ | $831 \pm 400$ | $1333 \pm 558$ | $945 \pm 652$ | $1352 \pm 273$ | $2004 \pm 106$ |
| HalfCheetah | 0 | $1210 \pm 391$ | $1278 \pm 373$ | $1235 \pm 317$ | $1655 \pm 296$ | $1728 \pm 179$ | $1601 \pm 241$ |
| | 1.1 | $434 \pm 415$ | $1055 \pm 265$ | $914 \pm 427$ | $967 \pm 365$ | $1275 \pm 42$ | $1330 \pm 297$ |
| | 1.2 | $1167 \pm 491$ | $988 \pm 446$ | $948 \pm 476$ | $1679 \pm 241$ | $1441 \pm 254$ | $1466 \pm 265$ |
| | 1.3 | $\mathbf{1506 \pm 552}$ | $\mathbf{1442 \pm 588}$ | $830 \pm 611$ | $\mathbf{2097 \pm 213}$ | $\mathbf{2081 \pm 175}$ | $1408 \pm 68$ |
| | 1.4 | $379 \pm 555$ | $1224 \pm 412$ | $\mathbf{1302 \pm 300}$ | $1187 \pm 305$ | $1534 \pm 40$ | $\mathbf{1659 \pm 107}$ |
| | 1.5 | $257 \pm 442$ | $527 \pm 727$ | $780 \pm 593$ | $755 \pm 542$ | $1480 \pm 646$ | $1550 \pm 92$ |

**Threshold Selection**   In our proposed method, we can control the magnitude of policy uniqueness flexibly by adjusting the constraint threshold $r_0$. Choosing different thresholds will lead to different policy behaviors. Concretely., a larger threshold may drive the agent to perform more differently while smaller threshold imposes a lighter constraint on the behavior of the agent. Intuitively, a larger threshold will lead to relatively poor performance for the learning algorithm is less likely to find a feasible solution to Eq.(7).

Besides, we do not use constraints in the form of Eq.(7) as we need not force every single action of a new agent to be different from others. Instead, we are more care about the long term differences. Therefore, we use the cumulative uniqueness as constraints,

$$
\max_{\theta \in \Theta} \quad \mathbb{E}_{\tau \sim \theta}[g_{\text{task}}],
$$
$$
s.t. \quad \sum_{t=0}^{t=\tau}(r_{\text{int},t} - r_0) \geq 0, \forall \tau = 1, 2, ..., T,
$$

We test our method with different choices of threshold values. The performance of agents under different thresholds are shown in Fig. 9 and more detailed analysis of their success rate is presented in Table. 2.

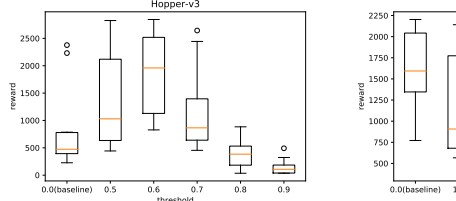 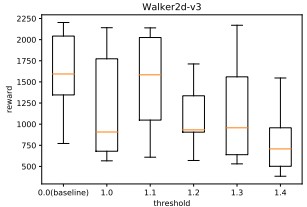 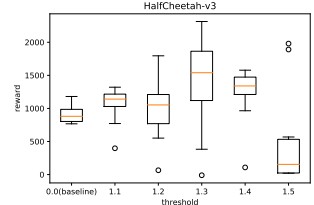

Figure 9: The performance our methods under different threshold selection, in the Hopper, Walker and HalfCheetah environments. the results are collected from 10 learned policies.

# E    MORE QUALITATIVE RESULTS

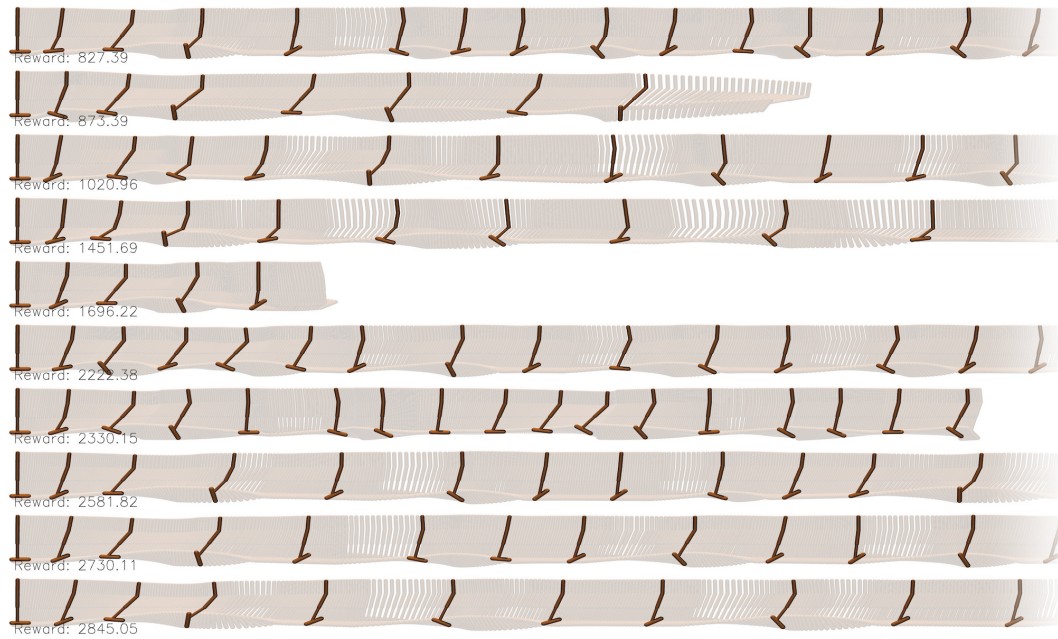

Figure 10: The visualization of policy behaviors of agents trained by our method in Hopper-v3 environment. Agents learn to jump with different strides.

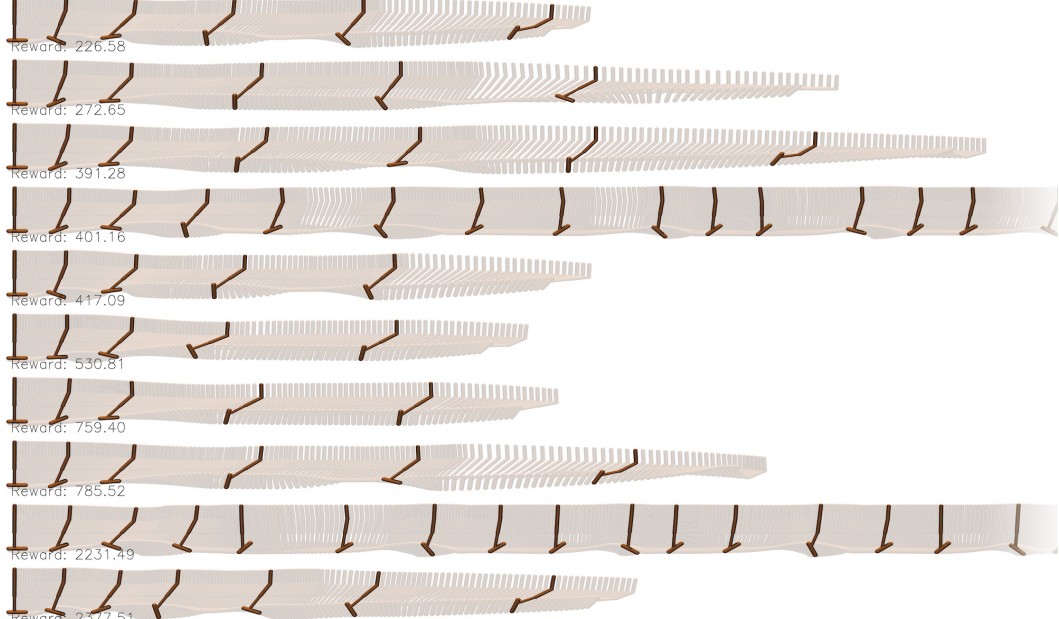

Figure 11: The visualization of policy behaviors of agents trained by PPO in Hopper-v3 environment. Most agents learn a policy that can be described as *Jump as far as possible and fall down*, leading to relative poor performance.

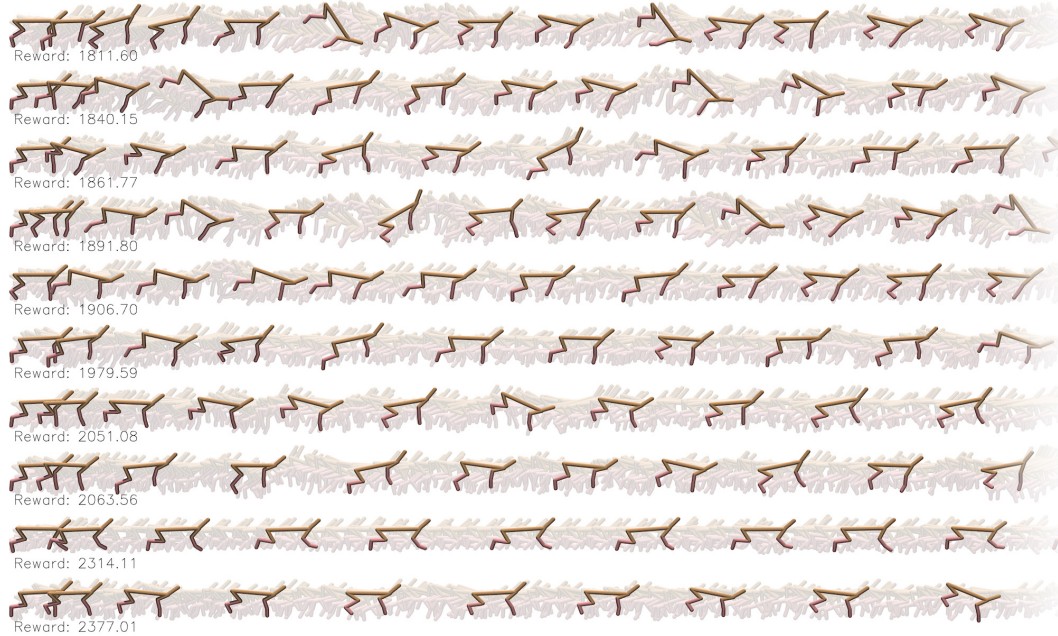

Figure 12: The visualization of policy behaviors of agents trained by our method in HalfCheetah-v3 environment. Our agents run much faster compared to PPO agents and at the mean time several patterns of motion have emerged.

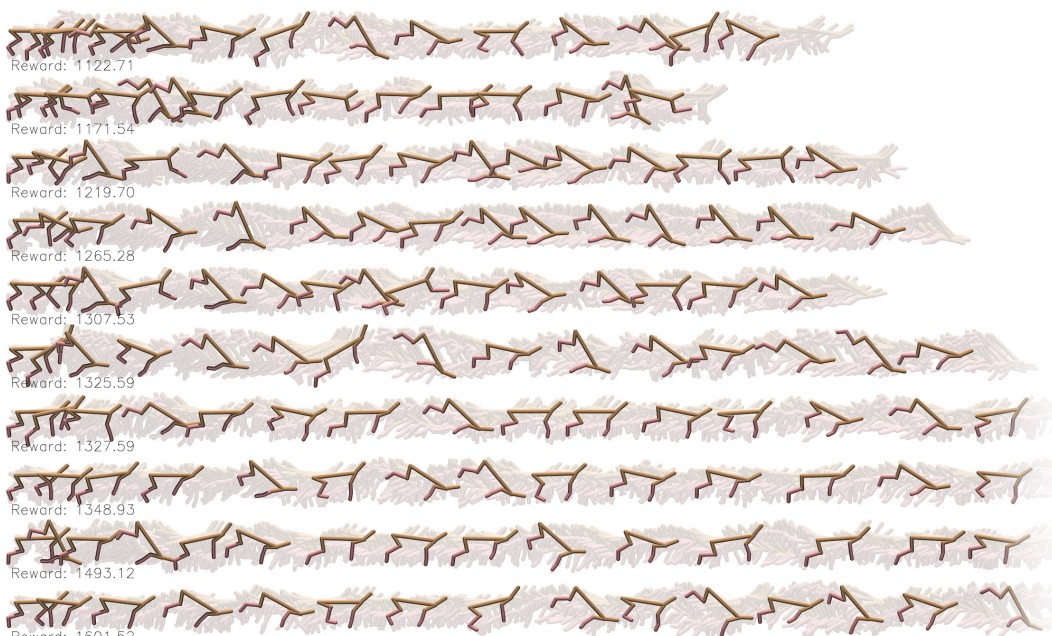

Figure 13: The visualization of policy behaviors of agents trained by PPO in HalfCheetah-v3 environment. Since we only draw fixed number of frames in each line, in the limited time steps the PPO agents can not run enough distance to leave the range of our drawing, which shows that our agents run much faster.

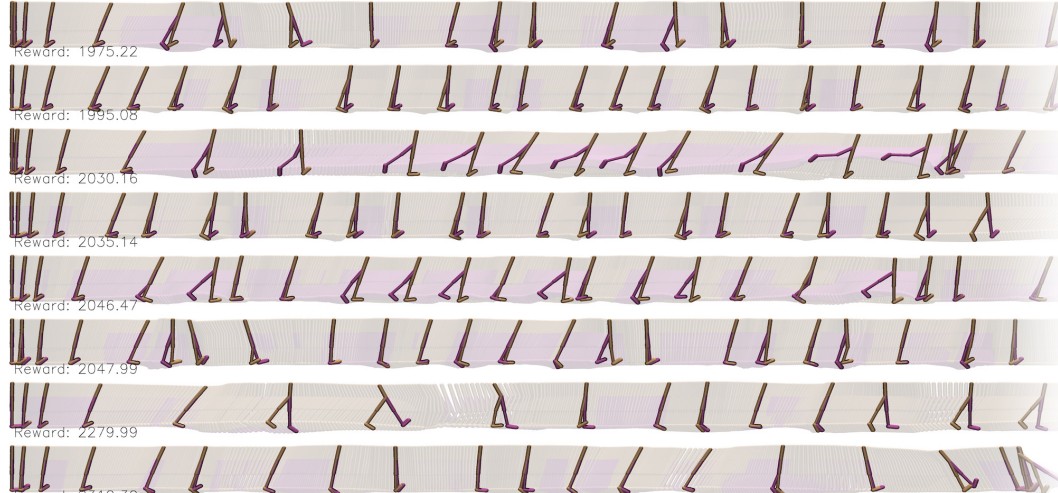

Figure 14: The visualization of policy behaviors of agents trained by our method in Walker2d-v3 environment. Instead of bouncing at the ground using both legs, our agents learns to use both legs to step forward.

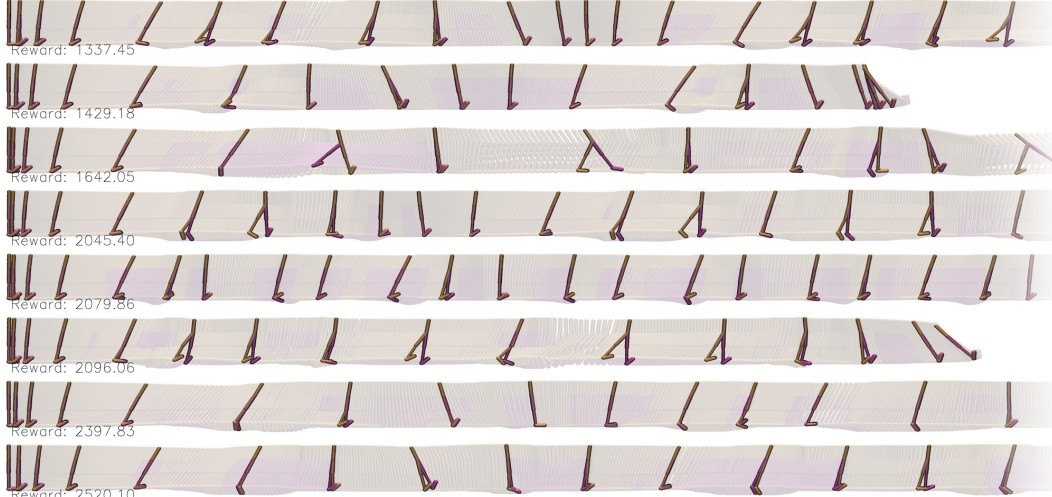

Figure 15: The visualization of policy behaviors of agents trained by PPO in Walker2d-v3 environment. Most of the PPO agents only learn to use both legs to support the body and jump forward.

## F  IMPLEMENTATION OF EQ.(7)

We do not use constraints in the form of Eq.(7) as we need not force every single action of a new agent to be different from others. Instead, we are more care about the long term differences. Therefore, we use the cumulative uniqueness as constraints. Moreover, the constraints can be applied after the first $t_S$ timesteps (e.g. $t_S = 20$) for the consideration of similar starting sequences.

$$
\max_{\theta \in \Theta} \quad \mathbb{E}_{\tau \sim \theta}[g_{\text{task}}],
$$
$$
s.t. \quad \textstyle\sum_{t=t_S}^{t=\tau}(r_{\text{int},t} - r_0) \geq 0, \tau = S, ..., T,
$$

$$(9)$$

# G RELATION BETWEEN DIFFERENT APPROACHES AND CONSTRAINED OPTIMIZATION METHODS

We note here, the WSR, TNB and IPD methods correspond to three approaches in constrained optimization problem. For simplicity, we consider Eq.(9) with a more concise notion $g_{\text{int},t} - g_{0,t} \geq 0$, where $g_{\text{int},t} = \sum_{t=0}^{t} r_{\text{int},t}$, i.e.,

$$
\begin{aligned}
\max_{\theta \in \Theta} \quad & f(\theta) = \mathbb{E}_{\tau \sim \theta}[g_{\text{task}}] \\
s.t. \quad & g_t(\theta) = g_{\text{int},t} - g_{0,t} \geq 0, t = 1, 2, ..., T
\end{aligned}
\tag{10}
$$

As the optimization of policy is based on batches of trajectory samples and is implemented with stochastic gradient descent, Eq.(10) can be further simplified as:

$$
\begin{aligned}
\max_{\theta \in \Theta} \quad & f(\theta) = \mathbb{E}_{\tau \sim \theta}[g_{\text{task}}] \\
s.t. \quad & g(\theta) = \overline{g}_t(\theta) \geq 0
\end{aligned}
\tag{11}
$$

where $\overline{g}_t(\theta)$ denotes the average over a trajectory.

**WSR: Penalty Method**   The Penalty Method considers the constraints of Eq.(11) by putting constraint $g(\theta)$ into a penalty term, and then solve the unconstrained problem

$$
\max_{\theta \in \Theta} \quad f(\theta) + \frac{1-\alpha}{\alpha} \min\{g(\theta), 0\}
\tag{12}
$$

using an iterative manner, and the limit when $\alpha \to 0$ lead to the solution of the primal constrained problem. As an approximation, WSR choose a fixed weight term $\alpha$, and use the gradient of $\nabla_\theta f + \frac{1-\alpha}{\alpha}\nabla_\theta g$ instead of $\nabla_\theta f + \frac{1-\alpha}{\alpha}\nabla_\theta \min\{g(\theta), 0\}$, thus the final solution will intensely rely on the selection of $\alpha$.

**TNB: Feasible Direction Method**   The Taylor series of $g(\theta)$ at point $\bar{\theta}$ is

$$
g(\bar{\theta} + \lambda\vec{p}) = g(\bar{\theta}) + \nabla_\theta g(\bar{\theta})^{\text{T}}\lambda\vec{p} + O(||\lambda\vec{p}||)
\tag{13}
$$

The Feasible Direction Method (FDM) considers the constraints of Eq.(11) by first finding a direction $\vec{p}$ satisfies

$$
\begin{aligned}
& \nabla_\theta f^{\text{T}} \cdot \vec{p} > 0 \\
& \nabla_\theta g^{\text{T}} \cdot \vec{p} > 0 \quad \text{if} \quad g = 0
\end{aligned}
\tag{14}
$$

so that for small $\lambda$, we have

$$
g(\bar{\theta} + \lambda\vec{p}) = g(\bar{\theta}) + \lambda\nabla_\theta g(\bar{\theta})^{\text{T}}\vec{p} > g(\bar{\theta}) = 0 \quad \text{if} \quad g(\bar{\theta}) = 0
\tag{15}
$$

and

$$
g(\bar{\theta} + \lambda\vec{p}) = g(\bar{\theta}) + \lambda\nabla_\theta g(\bar{\theta})^{\text{T}}\vec{p} > 0 \quad \text{if} \quad g(\bar{\theta}) > 0
\tag{16}
$$

The TNB method, by using the bisector of gradients $\nabla_\theta f$ and $\nabla_\theta g$, select $\vec{p}$ to be

$$
\vec{p} = \begin{cases} \nabla_\theta f + \frac{|\nabla_\theta f|}{|\nabla_\theta g|}\nabla_\theta g \cdot \cos\left(\nabla_\theta f, \nabla_\theta g\right) & \text{if } \cos\left(\nabla_\theta f, \nabla_\theta g\right) \leq 0 \\ \nabla_\theta f + \frac{|\nabla_\theta f|}{|\nabla_\theta g|}\nabla_\theta g & \text{if } \cos\left(\nabla_\theta f, \nabla_\theta g\right) > 0 \end{cases}
\tag{17}
$$

Clearly, Eq.(17) satisfies Eq.(14), but it is more strict than Eq.(14) as the $\nabla_\theta g$ term always exists during the optimization of TNB. In TNB, the learning stride is fixed to be $\frac{|\nabla_\theta f| + |\nabla_\theta g|}{2}$, leading to problem when $\nabla_\theta f \to 0$, which shows the final optimization result will heavily rely on the selection of $g$. i.e., the shape of $g$ is crucial for the success of TNB.

**IPD: Interior Point Methods (IPMs)**   In vanilla IPMs, the constrained optimization problem in Eq.(11) is solved by reforming it to an unconstrained form with an additional barrier term $\alpha \frac{1}{g(\theta)}$ in the objective as

$$\max_{\theta \in \Theta} \quad f(\theta) + \alpha \frac{1}{g(\theta)} \tag{18}$$

or use the barrier term of $-\alpha \log g(\theta)$ instead:

$$\max_{\theta \in \Theta} \quad f(\theta) - \alpha \log g(\theta) \tag{19}$$

where $\alpha$, the barrier factor, is a small positive number. As $\alpha$ is small, the barrier term will introduce only minuscule influence on the objective. On the other hand, when $\theta$ get closer to the barrier, the objective will increase fast. It is clear that the solution of the objective with barrier term will get closer to the primal objective as $\alpha$ getting smaller. Thus in practice, such methods will choose a sequence of $\{\alpha_i\}$ such that $0 < \alpha_i < \alpha_{k+1}$ and $\alpha_i \to 0$ as $k \to \infty$ The limit of Eq.(18), Eq.(19) when $\alpha \to 0$ then leads to the solution of Eq.(11). The work of Conn et al. (1997); Wright (2001) provide proofs of the convergence.

Directly applying this method is computationally challenging and numerically unstable, especially when $\alpha$ is small. A more natural way can be used: since the learning process is based on sampled transitions, we can simply bound the collected transitions in the feasible region by permitting previous trained $M$ policies $\theta_i \in \Theta_{\text{ref}}, i = 1, 2, ..., M$ sending termination signals during the training process of new agents. In other words, we implicitly bound the feasible region by terminating any new agent that steps outside it.

Consequently, during the training process, all valid samples we collected are inside the feasible region, which means these samples are less likely to appear in previously trained policies. At the end of the training, we then naturally obtain a new policy that has sufficient uniqueness. In this way, we no longer need to consider the trade-off problem between intrinsic and extrinsic rewards deliberately. The learning process of our method is thus more robust and no longer suffer from objective inconsistency. Algorithm.1 shows the pseudo code of IPD based on PPO, where the blue lines show the addition to primal PPO algorithm.

---

**Algorithm 1** IPD with PPO, Actor-Critic Style

---

**Require**
- a behavior policy $\theta_{\text{old}}$
- a set of previous policies $\{_j\}, j = 1, 2, ..., M$
- a uniqueness metric $U(\theta, \{\theta_j\}|\rho) = U(\theta, \{\theta_j\}|\tau) = \min_{\theta_j} \overline{D}_{TV}^{\tau}(\theta, \theta_j)$
- a uniqueness threshold $r_0$, starting point $t_S$

Initialize $\theta_{\text{old}}$
**for** iteration $= 1, 2, ...$ **do**
  **for** actor $= 1, 2, ..., N$ **do**
    **for** t $= 1, 2, ..., T$ **do**
      Run policy $\theta_{\text{old}}$ in environment, get trajectory $\tau$
      **if** $U(\theta_{\text{old}}, \{\theta_j\}|\tau) - r_0 < 0$, AND $t > t_S$ **then**
        done = True
      **end if**
      **if** done **then**
        break
      **end if**
    **end for**
    Compute advantage estimates $\hat{A}_1, ..., \hat{A}_T$
  **end for**
  Optimize surrogate $\mathcal{L}^{\text{CLIP}}$ w.r.t. $\theta$, with $K$ epochs and minibatch size $M \leq NT$
  $\theta_{old} \leftarrow \theta$
**end for**

---

