# OpenReview forum: "Learning with Social Influence through  Interior Policy Differentiation"
_ICLR.cc/2020/Conference — Reject_

### Official Review · AnonReviewer1 · 2019-10-23
**Official Blind Review #1**

**Rating:** 3

**Review:**

This paper proposes a new method for learning diverse policies in RL environments, with the ultimate goal of increasing reward. The paper develops a novel method, called interior policy differentiation (IPD), that constrains trained policy to be sufficiently different from one another. They test on 3 Mujoco domains, showing improved diversity in all of them and improved performance in 2 of them.

Overall, this paper is very well executed. The explanation of the method is thorough, and the paper is well-written and polished. I like the idea of enforcing a constraint on the policy diversity via manipulating the transitions that the agents can learn from.  The experiments section compares to two other methods of increasing policy diversity and IPD outperforms both of them. I think this is a solid contribution to the literature on improving policy diversity.

That being said, I have some concerns about the paper:
1) The motivation for explicitly encouraging diverse policies is a bit confusing, and isn’t very convincing. The paper draws inspiration from social influence in animal society, and say they formulate social influence in RL. First, the term social influence already has an established meaning in RL (see e.g. Jaques et al. (2018)) and refers to agents explicitly influencing others in a causal way in a multi-agent environment. Second, I think calling policy diversity a form of social influence is a bit of a stretch (and anthropomorphizes the agents unnecessarily). I think the paper should scrap the ‘social influence’ angle and instead frame it as ‘increasing policy diversity’.

The paper also motivates itself in comparison to Heess et al. (2017), which uses a set of environments to get diverse policies. However, the goal of these works are different: in Heess et al., the goal is to train agents that can exhibit complex behaviours in relatively simple environments (the focus is more on complexity of behaviours vs. the fact that agents in the same environment learn diverse policies). In this work, the goal is not to develop any more complex policies, but to have different agents on the same task learn diverse policies (and since the experiments are in Mujoco, the degree of diversity is limited). Thus, while the works are related, I don’t think the Heess et al. paper is good motivation for this work.

I think the primary motivation that makes sense for explicitly encouraging diversity is to improve final performance on the task. Thus, I think it would be best for the paper to clarify the introduction by focusing on this. The paper could also give some reasons why having diverse policies is inherently a good thing (maybe for some applications with humans-in-the-loop it could be helpful?), but currently this is absent.

2) Given that improving the final reward of an RL agent is the main goal, it’s not clear that the experiments (in 3 simple Mujoco settings) are enough to show this reliably. Specifically, it’s unclear whether encouraging diversity in this way will generalize to more complex tasks or domains (e.g. tasks in Mujoco with sparser reward, or environments with larger state spaces). It is possible that the success of the technique is most prevalent when there is only a small observation space.

3) I’d like to see more discussion / analysis of *why* we’d expect diverse policies to lead to better rewards. In work on intrinsic motivation / curiosity for better exploration, it’s clear that encouraging agents to visit unseen states will lead to a better exploration of the state space, and thus will make them more likely to stumble upon rare rewards. But is this also true for policy diversity? Currently, the paper speculates that encouraging diversity could help agents not all fall into the same failure mode. But I could also imagine that it could lead agents to avoid a successful strategy that another agent learned. For example, if a certain sequence of moves is necessary at the beginning to avoid termination, the first agent could find this sequence of moves, but the other agents might avoid this sequence for the sake of diversity (depending on the threshold). Does something like this happen in practice? In my opinion, the environments considered aren’t rich enough to know.

4) There are also some inconsistencies in results section. Specifically:
- There seems to be a disagreement between the results in Table 1 (which uses 10 peers) and the ablation over number of peers in Figure 4, which shows that the performance with 10 agents is roughly the same as it is with 1 agent (and overall shows little positive trend between the number of peers and performance).
- If ‘success rate’ means ‘percentage of time beating average PPO policy’, why does PPO sometimes get 100% in Table 1?

Given the concerns above, I’d assess the paper as being borderline for accept. I’m currently erring on the side of rejection, but I’d consider changing my score if some of the above points are addressed.


Smaller concerns and questions:
- There are a couple of instances where I found the claims of the paper with respect to related work to be over-stated. For example:
‘Yet designing a complex environment requires a huge amount of manual efforts’ -> not necessarily. There is an initial engineering overhead, but it’s possible to generate environments programmatically with different properties, resulting in different agent behaviours.
Also:
On the Task-Novelty Bisector method of (Zhang et al., 2019): ‘the foundation of such joint optimization is not solid’. This is given without any explanation --- how is it not solid?

- In the TNB and WSR implementation, what metric is being used? Is it the same as is defined in Section 3?

- It would be nice to have some videos of the agents behavior to be able to more easily assess the learned policy diversity.


Small fixes:
‘and similar results can be get’ -> and get similar results.


**Experience Assessment:**

I have published one or two papers in this area.

**Review Assessment: Checking Correctness Of Derivations And Theory:**

I assessed the sensibility of the derivations and theory.

**Review Assessment: Checking Correctness Of Experiments:**

I carefully checked the experiments.

**Review Assessment: Thoroughness In Paper Reading:**

I read the paper at least twice and used my best judgement in assessing the paper.

---

> ### Author Response · Authors · 2019-11-07
> **Re: Official Blind Review #1**
>
> Thank you for the insightful comments.
>
> 1) The Motivation for Policy Diversity:
> The motivation of our work is to find different policies for the same task with high efficiency.
> There are applications where the diversity of policies is useful. 1. It may help empower agents with different styles or characters, which is especially useful in traffic simulations and video games where each instance is expected to behave differently. 2. Sometimes different policies themselves are important: applying diversity seeking methods in quantitative trading may help to discover different trading strategies (e.g., momentum and reversion), with each unique policy act as an *Alpha* to earn extra profit. 3. The performance w.r.t. a previously assigned reward function only represents the reward function’s preference. While learning diverse policies can help the agent to develop different strategies and become less reliant on the reward function. We will clarify the motivation in the Intro. section.
>
> 2) The Inspiration from Heess et al. 2017:
> Retrospect the success of Heess et al. 2017, their agent learns to move forward with different poses according to the obstacles in front of them. Those running, jumping and climbing policies can be regarded as diverse policies for the task *moving forward*. The forward bonus is the only reward they utilized. Attributing their success to Darwinism (i.e., the survival of the fittest), we are inspired to generate different policies with different termination signals, and we further proposed to couple the uniqueness constraints with the environment, eventually forming the IPD.
> Regarding the traditional RL learning paradigm as Darwinism, our proposed learning paradigm enables agents not only interact with the environment but also with its peers, which is the reason we introduce the concept of social uniqueness in our work. We will reduce the usage of social influence in the revision.
>
> 3) More on Performance Improvement:
> Given the motivation to find diverse policies in (1), we demonstrate our proposed method of generating diverse policies efficiently rather than generating better-performed policies. i.e., we embrace all diversities including the poor-performed ones and regard them equal as peers. When the peers perform well (e.g. 10 policies trained with PPO in Walker), a new agent trying to be different from its peers will perform poorly. On the contrary, when the peers perform poor in general (e.g. 10 policies trained with PPO in Hopper and HalfCheetah), a new agent trying not to resemble its peers will tend to perform better and result in performance improvement. As our paper focuses on learning different policies, we only regard the performance-boosting in some cases as byproducts. Combining the learned diverse policies and enforcing different policies to have better performance are promising topics in future work.
>
> 4) For the Claim and Other Concerns:
> We have revised some of the mentioned parts of the paper.
> - As for the *inconsistency*: in our main results (Table 1. and Fig.3 ), all of the experiments start with 10 PPO policies as peers. But in the results shown in Fig.4, the experiments started with 1 PPO policy, as follows: the first policy is trained with PPO, the second policy is trained with IPD to be different from its first peer, and so on. And we repeat the WHOLE PROCESS 5 times to get average results. Consequently, as there are more poor policies in the 10 PPO peers for Hopper and HalfCheetah, the performance of IPD surpasses PPO in those two environments.
> - In Heess et al. 2017, the diversity of learned moving strategies is limited by the number of different kinds of terrains and obstacles, i.e., their agent can not learn to jump forward without an obstacle in front of it. Thus, the *diversity* (if we regard their different skills as policies, although this is not true for such skills are combined in one policy and only get triggered when facing corresponding terrains or obstacles) is limited by the diversity of environment and is hard to produce different policies in batch.
> - In TNB, the final performance will rely on the scale of intrinsic reward, and we provide a detailed analysis of WSR, TNB in Appendix G to analyze their deficiencies. With the analysis, TNB can be revised with a condition to avoid too much optimization in the direction of $r_{novel}$ or $L_{novel}$.
> - The success rate in Table 1 is defined as *surpasses the baseline during training process* so that PPO sometimes gets 100% for they always achieve similar performances during training.
> - In TNB and WSR, the same metric is used
> - We will try our method on more environments as well as in loosed environments that permit the agent to survive in more states.

---

> > ### Comment · AnonReviewer1 · 2019-11-11
> > **Response to authors**
> >
> > Thank you for the rebuttal. After consideration, I'm keeping my score the same, as I am still not convinced by the utility of the policy diversity argument. I'd encourage the authors to explore their method in a concrete setting where this has demonstrable advantages.

---

### Official Review · AnonReviewer2 · 2019-10-24
**Official Blind Review #2**

**Rating:** 3

**Review:**

This paper proposes a new way to incentivize diverse policy learning in RL agents: the key idea is that each agent receives an implicit negative reward (in the form of an early episode termination signal) from previous agents when an episode begins to resemble prior agents too much (as measured by the total variational distance measure between the two policy outputs).

Results on three Mujoco tasks are mixed: when PPO is combined with the proposed objective for training diverse policies, it results in very strong performance boosts on Hopper and HalfCheetah, but falls significantly short of standard PPO on Walker 2D. I would have liked to see a deeper analysis of what makes the approach work in some environments and not in others.

Experimental comparisons in the paper are only against alternative approaches to optimize the same diversity objective as the proposed approach (with weighted sum of rewards (WSR) or task novel bisection(TNB)). Given that this notion of diversity is itself being claimed as a contribution, I would expect to see comparisons against prior methods, such as in DIAYN. There are other methods that have been proposed before in similar spirit to induce diversity in the policies learned. Aside from the evolutionary approaches covered in related work, within RL too, there have been methods such as the max-entropy method proposed in Eysenbach et al, "Diversity is All You Need...". These methods, evolutionary and RL, could be compared against to make a more convincing experimental case for the proposed approach.

The experimental setting is also not fully clear to me: throughout experiments, are the diversity methods being evaluated for the average performance over all the policies learned in sequence to be different from prior policies? Or only the performance of the last policy? Related, I would be curious to know, if K policies are trained, the reward vs the training order k of the K policies. This is close to, but not identical to the study in Fig 4, to my understanding.

Aside from the above points being unclear, the paper in general could overall be better presented. While I am not an expert in this area, I would still expect to be able to understand and evaluate the paper better than I did.
- Sec 3.1 makes a big deal of metric distance, but never quite explains how this is key to the method.
- The exact baselines used in experiments are unhelpfully labeled "TNB" (with no nearby expansion) and "weighted sum of rewards (WSR)", with further description moved to appendix. In general, there are a few too many references to appendices.
- The results in Fig 2 are difficult to assess for diversity, and this is also true for the video in the authors' comment.
- There is an odd leap in the paper above Eq 7, where it claims that "social uniqueness motivates people in passive ways", which therefore suggests that "it plays more like a constraint than an additional target".
- Sec 5.1 at one point points to Table 1 for "detailed comparison on task related rewards" but says nothing about any important conclusions from the table.
- There are grammar errors throughout.

**Experience Assessment:**

I do not know much about this area.

**Review Assessment: Checking Correctness Of Derivations And Theory:**

I did not assess the derivations or theory.

**Review Assessment: Checking Correctness Of Experiments:**

I assessed the sensibility of the experiments.

**Review Assessment: Thoroughness In Paper Reading:**

I read the paper at least twice and used my best judgement in assessing the paper.

---

> ### Author Response · Authors · 2019-11-07
> **Re: Official Blind Review #2**
>
> Thank you for the insightful comments.
>
> 1) The Motivation of IPD:
> In previous diversity seeking approaches, the diversity reward or novelty reward is always considered as an extra term of the learning objective, leading to a deformation of the loss function landscape, and therefore deliberate justification is needed. To tackle such a problem, our proposed method draws key insight from the social uniqueness motivation of human society. Specifically, given the same primal task, people tend to reach the objective in different ways, which is defined as social uniqueness motivation in the psychology literature (Chan et al. 2012). Such uniqueness motivation is not an explicit objective but a constraint, providing inspiration to our work on how to avoid distorting the loss landscape. This is also the motivation we rewrite the problem from Eq.(6) to Eq.(7) in our work.
>
> 2) Relation with DIAYN, and Performance Analysis:
> The work of DIAYN can be regarded as a kind of curiosity-driven method that motivates an agent to explore more previous unseen states. While DIAYN categorizes ** different skills within a policy ** conditioned on the latent variable $Z$, our approach concentrates on increasing the differences among policies, i.e., different behaviors between policies.
> DIAYN, as a sort of meta-learner, learns skills in an unsupervised way and can be used as pre-training in various tasks to get performance-boosting ( thus in reward sparse DIAYN is especially useful).
> On the other hand, we demonstrate our proposed method of generating diverse policies efficiently rather than generating better-performed policies. i.e., we embrace all diversities including the poor-performed ones and regard them equal as peers. When the peers perform well (e.g. 10 policies trained with PPO in Walker), a new agent trying to be different from its peers might perform poorly. On the contrary, when the peers perform poor in general (e.g. 10 policies trained with PPO in Hopper and HalfCheetah), a new agent trying not to resemble its peers will tend to perform better and result in performance improvement.
>
> 3) Experimental Settings:
> We provide an algorithm box in Appendix G to make our method more clear.
> The results of main experiments (Table 1. and Fig.3 ) are executed as follows: first, we train 10 PPO policies, and then train another 10 policies with different methods, i.e., WSR, TNB or IPD separately. The performance (reward) shown in Table 1 are averaged over 10 policies in each method so that they correspond to the Y-axis of Fig.3.
> The experimental results in Fig.4 come as follows: the first policy is trained with PPO, the second policy is trained to be different from the former peer, and so on. And we repeat the WHOLE PROCESS 5 times to get averaged results (shown in Fig.4). The results in Fig.4 show great variance, and we interpret such variance from the reliance of later policy performance on its peers (When peers perform well, a new agent trying to be different from its peers will perform poorly...). Intuitively, as the number of peers increases, the new policy will get more constraints so that it will get harder to find a feasible policy, especially in simple tasks where diversity is limited by the environment (e.g. the Hopper). And we do observe a clear decrease in the curve for Hopper in Fig.4.
>
> 4) Details on Baselines
> Restricted by the page limitation, we put a detailed introduction to WSR and TNB into Appendices. We have updated some analysis of those methods, mainly on their correspondence between constrained optimization problems, in Appendix G. We will move more information into the main text in our revision.
>
> 5) On the Metric in Sec.3.1
> The metric is important for it enables fast and rigorous computation of differences between policies, i.e., it guarantees the self-consistency of the distance between policies. Moreover, it lays the foundation for Proposition 1, based on which we implement our algorithm with single trajectory estimation and further improve the learning efficiency (compared with sampling states from the whole state space directly).

---

### Official Review · AnonReviewer3 · 2019-10-26
**Official Blind Review #3**

**Rating:** 3

**Review:**

The paper presents a new algorithm for maximizing the diversity of different policies learned for a given task. The diversity is quantified using a metric, where in this case the total variation is used. A policy is different from a set of other policy if its minimum distance to all the other policies is high. The authors formulate a new constraint optimization problem where the diversity to previous policies is lower bounded in order to avoid a tedious search for combining task reward and diversity reward. The algorithm is evaluated on different Mojoco locomotion tasks.

Positive Points:
- The idea of maximizing the minimum total variation is novel and interesting
- The approach seems to work better than current SOTA approaches for generating diverse behavior

Negative Points:
- The paper needs to be improved in terms of writing as in particular some of the main parts of the algorithm are unclear
- The definition of Eq 7 does not make too much sense to me (see below)
- The results have high variance and some conclusion drawn from it are hard to verify given the plots

More comments:
- Eq 7 does not seem to be a very good choice to me. Why does the total variation needs to be different at *every* time step? We can certainly generate very diverse behavior even if the policy is exactly the same for some states. It could even be the case that for some states, only one action does not lead to a failure. In this case, Eq 7 would completely fail to produce any valid policy (?)
- In general the writing is clear, it gets however quite unclear for the main part of the algorithm (after Eq. 7). It is unclear how equation 8 is obtained and why the limit of alpha going to 0 should lead to the same solution as Eq 7 (if alpha is 0 than it should be the same as optimizing just the task reward??). While this might be obvious for experts of the interior point method, it needs to be explained in much more detail in this paper. I think it is always a good strategy to make a paper self-contained, in particular for the main parts of the algorithm.
- Also the termination mechanism needs to be much better explained. What reward is given in this case? The current formulation sounds quite heuristic to me, but maybe a better explanation can fix that.
- while Fig 3 shows a clear advantage of the method, the section about better policy discovery would need better data to verify their claims. Fig 4 shows very noisy results and while for the hopper there might be a clear improvement of performance for number of policies > 2, this does not seem to be very significant for half cheetah. Given the amount of noise in the results many more trials would be needed to really make such statements.


**Experience Assessment:**

I have published one or two papers in this area.

**Review Assessment: Checking Correctness Of Derivations And Theory:**

I carefully checked the derivations and theory.

**Review Assessment: Checking Correctness Of Experiments:**

I carefully checked the experiments.

**Review Assessment: Thoroughness In Paper Reading:**

I read the paper thoroughly.

---

> ### Author Response · Authors · 2019-11-07
> **Re: Official Blind Review #3**
>
> Thank you for the insightful comments.
>
> We will try to make the second contribution of our work more clear in the revision. Specifically, the IPD method for policy differentiation. The IPD provides a general framework that can be applied whenever there is more than one objective in RL to optimize, e.g., learning with demonstrations (where there is an extra behavior cloning loss), curiosity-driven learning (where there is an extra curiosity bonus), etc.
> In short, it considers the extra loss or bonus in the sample-collection process, executed by early termination, so that policies trained with such samples will naturally satisfy the constraints.
>
> 1) Eq.7:
> Yes. In practice, we should not expect a new agent to be different from others at every timestep, where a moving average is utilized. In fact, our uniqueness metric is based on sampled trajectories (and Proposition 1 shows we can use a single trajectory to get unbiased estimation). Thus, $r_{int}$ is kind of a moving average naturally. We have revised Eq.7 to be more precise and try to avoid being misled. We also provide more implementation details in Appendix D (paragraph: Threshold Selection) and Appendix F.
>
> 2) Algorithm and the IPD Method:
> We provide an algorithm box in Appendix G to make our method more clear. Inspired by the Interior Point Methods, our proposed method tackles the uniqueness credit assignment problem in a quite different but natural way. i.e., we need not assign reward but only need to send termination signals to the agent during training when the constraints are broken. Moreover, based on our proposed uniqueness metric, the algorithm is easy to implement and will be easy to apply to any other prevailing RL algorithms.
> In Appendix G, we also provide an analysis of all the three methods (WSR, TNB, and IPD) on their relationship between constrained optimization problems, namely the WSR—Penalty Method, TNB—Feasible Direction Method, and IPD—Interior Point Method.
>
> 3) Variance and Experimental Settings
> As our method is proposed to seek uniqueness (diversity), the randomness in the learning process is a must. Moreover, as we train policies sequentially, the randomness will be accumulated, leading to a high variance in Fig. 4.
> The experimental results Fig.4 show comes as follows: the first policy is trained with PPO, the second policy is trained to be different from the former peer, and so on. And we repeat the WHOLE PROCESS 5 times to get averaged results (shown in Fig.4). Intuitively, suppose the first 5 policies are poor, the 6th policy will have a larger chance to perform better (because more poor policies are considered as *peers it should not be similar with*), and vice versa.
> In our main results (Table 1. and Fig.3 ), the experiments start with 10 PPO policies as peers. Consequently, if there are more poor policies in the 10 PPO peers (e.g., in Hopper and HalfCheetah), the performance of IPD will surpass PPO. Otherwise, if the 10 PPO peers perform quite good, IPD might only able to find poorly performed policies to be different from the good ones. As our paper focuses on learning different policies, we only regard the performance-boosting in some cases as byproducts. And combining the learned diverse policies and enforcing different policies to have better performance are promising topics in future work.

---

### Author Response · Authors · 2019-10-06
**Demo Video**

Dear reviewers and general audience,

Please download the demo video following this dropbox link: https://www.dropbox.com/s/rrs08dicidcim2l/ICLR20.mp4

---

### Decision · Program_Chairs · 2019-12-19

**Decision:**

Reject

**Comment:**

The paper proposes a mechanism for obtaining diverse policies for solving a task by posing it as a multi-agent problem, and incentivizing the agents to be different from each other via maximizing total variation.

The reviewers agreed that this is an interesting idea, but had issues with the placement and exact motivations -- precisely what kind of diversity is the work after, why, and what accordingly related approaches does it need to be compared to.
Some reviewers also found the technical and exposition clarity to be lacking.

Given the consensus, I recommend rejection at this time, but encourage the authors to take the reviewers' feedback into account and resubmit to another venue.